# An Energy Saving Road Sweeper Using Deep Vision for Garbage Detection

**Luca Donati [1,*]**, **Tomaso Fontanini [1]**, **Fabrizio Tagliaferri [2]** and **Andrea Prati [1]**

[1]  IMP Lab, Department of Engineering and Architecture, University of Parma, 43121 Parma, Italy; tomaso.fontanini@studenti.unipr.it (T.F.); andrea.prati@unipr.it (A.P.)

[2]  Dulevo International S.p.a., Via Giovannino Guareschi 1, 43012 Fontanellato (PR), Italy; f.tagliaferri@dulevo.com

*  Correspondence: luca.donati@unipr.it; Tel.: +39-052-1905-785

**Abstract:** Road sweepers are ubiquitous machines that help preserve our cities cleanliness and health by collecting road garbage and sweeping out dirt from our streets and sidewalks. They are often very mechanical instruments, needing to operate in harsh conditions dealing with all sorts of abandoned trash and natural garbage. They are usually composed of rotating brushes, collector belts and bins, and sometimes water or air streams. All of these mechanical tools are usually high in power demand and strongly subject to wear and tear. Moreover, due to the simple working logic often implied by these cleaning machines, these tools work in an "always on"/"max power" state, and any further regulation is left to the pilot. Therefore, adding artificial intelligence able to correctly operate these tools in a semi-automatic way would be greatly beneficial. In this paper, we propose an automatic road garbage detection system, able to locate with great precision most types of road waste, and to correctly instruct a road sweeper in order to handle them. With this simple addition to an existing sweeper, we will be able to save more than 80% electrical power currently absorbed by the cleaning systems and reduce by the same amount brush weariness (prolonging their lifetime). This is done by choosing when to use the brushes and when not to, with how much strength, and where. The only hardware components needed by the system will be a camera and a PC board able to read the camera output (and communicate via CanBus). The software of the system will be mainly composed of a deep neural network for semantic segmentation of images, and a real-time software program to control the sweeper actuators with the appropriate timings. To prove the claimed results, we run extensive tests onboard of such a truck, as well as benchmark tests for accuracy, sensitivity, specificity and inference speed of the system.

**Keywords:** garbage detection; road sweeping; semantic segmentation; realtime neural networks

## 1. Introduction

Sweeping machines are a broad range of trucks tasked with the clean-up of most city streets and squares. They operate collecting garbage, gravel, trash, grass, dirt and leafs from the roads and the sidewalks. The main sweeping tools of such trucks are a variable number of rotating brushes, with the job of sweeping the road floor and "lifting" trash for the consecutive collection. A vacuum and/or conveyor belt will carry the collected garbage inside an internal bin for opportune disposal. These brushes and the belt have different settings, such as pressure on the floor, speed of rotation and speed of the belt, each tailored to a specific type of trash and, therefore, they need an accurate maneuvering for best performance. Moreover, these components are often very energy hungry, since cleaning very dirty road pavements may involve strong pressures and speeds. Besides the movement of the vehicle, these cleaning tools are the most energy-absorbing parts of these small trucks.

All of these systems are in fact powered by combustible fuel, or by an electric battery. In either case, the cleaning routine (driving the vehicle around the city) is performed by a human operator, and, as such, a typical usage scenario is a workday of eight hours of cleaning for these sweepers. Especially in the case of battery powered cleaners, this poses some constrains on the energy used by the vehicle, as every second of battery should be preserved in order to complete a daily routine without recharging/refueling the vehicle.

Another problem with these machines, operating in hard, real-life conditions from all over the world, is the wear-and-tear of its components. Many parts of these sweeper machines, especially the brushes, need to be replaced every month or so. This is both a huge cost and an additional source of waste material.

Fortunately, most of these energy efficiency problems can be solved by an accurate operation of the cleaning tools. The first power/weariness saving procedure is not cleaning where there is not any trash. Other power saving options are applying less brush pressure in case of "light" litter (e.g., leaves) as opposed to more difficult waste (e.g., bottles, mud).

Both of these proposals are unfortunately left as the responsibility to the truck pilot, who may be or may be not trained well for such operations, and would need to invest much of his/her attention to those fine regulation tasks, diverting it from other important tasks (e.g., driving). The usual consequence of such considerations is that often these machines are operating in an "always on"/"full cleaning" power setup, wasting energy, money and creating waste, noise and pollution.

Therefore, the need of an automated artificial intelligence system for these tasks arises. Having a computer automatically adjusting these settings would be valuable from both an economic and an ecological standpoint. The scope of this paper is to create an artificial intelligence system mounted on-board on the vehicle, able through a camera (eye) and a neural network (brain) to correctly identify each single piece of garbage encountered on the streets (leaves, paper, gravel, bottles, cigarette stubs, etc.). Moreover, the system will be able to convert the trash locations from the image coordinates to 3D real-world coordinates, and to collect all of them automatically through the truck actuators (brushes and vacuum).

Fortunately, nowadays cameras are quite cheap, performant and energy efficient, and the same can be said for embedded PC boards and GPUs. At the same time, deep vision technology has seen some mayor breakthroughs, and is now able to perform most visual recognition tasks with a human-like performance. The main challenge and contribution of this work is to verify if everything is ready to be put together in a real-time, cheap, energy saving system that can be implemented in a real truck.

The paper is structured as follows: Section 2 will highlight different aspects of our study and compare them to the most relevant previous works in the field; Section 3 will describe the full system for autonomous garbage detection; Section 4 will prove the system correctness and justify the choices presented in the main section; finally Sections 5 and 6 will draw conclusions about the work and propose interesting future developments.

## 2. Related Works

To the best of our knowledge, no article discusses a system for real-time garbage detection and collection using a full-sized road sweeper (such as the one in Figure 1). The most similar applications we found in the literature are related to the small, autonomous, cleaning robot scenario. These small (less than 50 cm wide) robots usually wander around indoor or outdoor environments, collecting trash and waste.

Other lines of work describe the trash collection problem at a higher level, outlining more the complexity of the problem and discussing the technology for its solution. These works do not focus on the vehicle actuators for waste collection, rather on the technology implied for garbage detection: computer vision and/or artificial intelligence applied on the data obtained from cameras, lasers, odorimeters, etc.

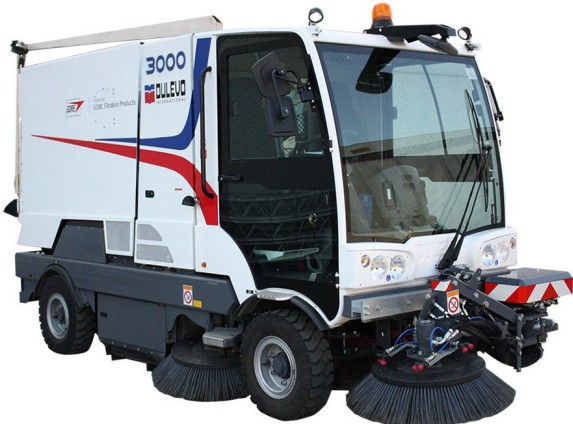

**Figure 1.** A typical three-brushes street sweeper (image courtesy of Dulevo International S.p.a.).

Another set of papers tackle the problem of detecting waste in underwater environments. While different from our scenario, some of the proposed approaches might be of interest.

We will discuss the most relevant past papers in the following subsections.

### 2.1. Trash Collecting Robots

The paper in [1] proposes a small robot for grass fields cleaning. It uses cameras and deep vision, and interestingly, an odometer, sonic sensor and a robot hand (manipulator). For garbage detection it uses SegNet [2] and ResNet [3]. While this is a valid approach, the task is quite different from ours: lower speeds, objects sizes are bigger (framed from a lower distance), "uniform" green background.

The paper in [4] instead proposes a laser guided robot for indoor cleaning. It uses a custom biological inspired network for navigation, but does not tackle trash detection directly.

The work in [5] is a small vision-guided robot that performs indoor floor cleaning with a custom-developed neural network. The paper proposes a combination of Convolutional Neural Networks [6] and Support Vector Machines [7]. Again, the domain is quite different from ours, mainly because of the indoor setup. The background for the robot to clean is a house/industry floor, and the speeds/camera components are completely different than in a road scenario.

The paper in [8] describes a small autonomous robot based on Arduino, that using only Dense Neural Networks [9] performs indoor cleaning. The remarks about being in a simplified domain are still valid, the test phase is performed on the floor of a house/industry, with mostly uniform backgrounds and big objects to be found, at low speeds.

The paper [10] outlines a Raspberry Pi-controlled robot able to run at 3–4 FPS to perform indoor waste detection and pickup. The system uses simple CNNs and is equipped with a robot arm for garbage collection. The reported accuracy is 90%, but the paper only shows a very limited set of indoor samples.

None of these robots are tested in on open roads or sidewalks and therefore cannot be directly compared with our vehicle, but they still provide some useful intuitions.

### 2.2. Indoor Trash Detection

The paper in [11] proposed a theoretical framework for trash detection. Interestingly, it does not make use of deep learning, but implies a combination of Histogram of Oriented Gradients [12] and Gray-Levels Co-occurence Matrix [13]. We argue that the setting of the tests is somewhat limited since it makes use of quite uniform indoor pictures.

The work in [14] is one of the most similar to the road sweeper scenario. The authors describe a small one-person vehicle that uses CNNs to detect water streaks in indoor locations and dry them.

While this is a quite controlled (industrial) environment as opposed to ours, still the work is interesting because it also make use of a Nvidia Jetson to perform real-time detection, similar to our approach. However, the indoor setup, and the task of detecting water streaks, are much easier conditions than our outdoor street/natural light domains.

### 2.3. Outdoor Trash Detection

The paper [15] highlights a clever system for smart cities monitoring. The idea is taking video streams around our cities to monitor the presence of abandoned garbage and illegal dumpsters. It uses Yolov3 [16] for garbage detection, and the waste to be found is typically big (trash bags, dumpsters, etc.). It focuses less on the single piece of waste, and more on the whole picture of the city cleanliness.

Similarly, the paper [17] makes use of Fast-RCNNs [18] to detect big clusters of garbage. Their dataset seems to be composed of outdoors pictures like the ones that could be taken from the Internet—i.e., it does not depict a real-world street scenario, with pictures taken from a running vehicle. Instead, the focus of this article, as opposed to ours, is the whole city management rather than garbage pickup, and so the proposed framework is tailored as such. Moreover, as most of these region-based networks proposals, the system can detect a big cluster of garbage, but is not developed to find small pieces of waste, like single cigarette stubs or leafs, as our system aims to do.

The recent paper in [19] presents a novel approach that performs an instance/semantic segmentation with a multi-level network based on multiple levels of U-Net [20] or FCN [21]. In particular, they integrate a coarse level network that proposes regions, a fine level (zoom) network that does per-pixel instance segmentation and also make use of a depth image to perform the full inference of their system. The article exhibits a comparison with a dataset (TACO) (with human produced garbage, indoor/outdoor, with people and different backgrounds, different framing distances) and also proposes a new dataset (called MJU-Waste), with depth images. Our first remark is the use of RGBD images, that, while interesting to study, is more of a toy problem than a real solution: it is tested in a limited indoor setup with a very near scene (one meter away from the camera) and a Depth of Field of a couple of meters. Those conditions are impossible to repeat in a moving street scenario with affordable cameras and sensors. The other concern is that the other dataset of test (TACO), while general enough, provides examples of trash that have two issues for our application: the represented garbage is usually quite ''big'' (bottles, boxes, plastic bags) and the represented garbage is totally human produced (again: bottles, bags, cans, etc.—i.e., no natural dirt). This dataset is interesting, but more limited than our setup, that contains small and/or natural waste such as gravel, soil, leaves, tree branches, cigarette stubs, etc. We also argue that human-produced trash is easier to detect than natural dirt, since human-produced trash is usually uniformly or brightly colored and has often text written over its surface.

Moreover, all of these methods do not tackle the problem of integrating a robot action after the detection of trash, while our method describes a completely integrated system within the cleaning vehicle.

### 2.4. Underwater Trash Detection

The paper in [22] performs underwater waste detection. A boat detects abandoned sea garbage using Yolo and R-CNNs [23]. Similarly, the work in [24] scans the seabed for litter using a Mask R-CNN [25].

In general, we argue that RCNN-like methods (Faster, Mask, Yolo) may be very useful for Instance Segmentation in general, but Semantic Segmentation with U-Net is better (and represents the most natural approach) when:

- The detection of single instances is not needed;
- Objects to be detected are many and/or small;
- Objects are not well bounded by a rectangle and/or overlapping.

We note that all of these concerns stand true for our application.

## 3. Description of The System

### 3.1. Camera Choice

The choice of a suitable camera is critical for a vehicle operating outside in the real world, and not in a controlled environment such as a laboratory or indoor environments. These vehicles are operating in snowy countries, desert areas, as well as rainy places. Light conditions may range from full summer light to night vision (with urban lights) or sunrise/sunset scenarios. Therefore, having a robust, dependable "eye" is mandatory for the correct work of the whole artificial intelligence system ("brain"). The camera to be used has thus some requirements that are important for the correct behavior of the solution:

- Low power consumption (powered by the battery even on fuel propelled vehicles);
- Adequate resolution (the higher the better, not under 2Megapixels);
- Sast, high bandwidth I/O with the PC (Gigabit Ethernet or better);
- Auto brightness;
- High sensor response (if, as for most of these vehicles, operating at night is a requisite);
- High Dynamic Range would be useful;
- Resistant to vibration and electromagnetic fields;
- Replaceable (as every industrial asset).

For most of these reasons, we chose a DALSA Genie Nano C1920 (@1920×1200), but many industrial cameras are supporting these requirements. The camera was placed inside the vehicle cockpit, avoiding temperature issues, damages for external hazards, and need for additional wipers for the lenses. The only downsides to be taken into account are windshield light reflections, which sometimes could glare the camera. The solution to this problem is placing the camera as close and as parallel as possible to the vehicle front glass, that in this case will act as an additional lens. Of course, the camera orientation will need to be finely tuned to the task it is performing.

Camera Optics and Placement

Road width varies across the world and from rural to urban areas. A good assumption for a single lane of a two-lane road is 3.5 m. Typical cars are narrower, and sweeper trucks make no exception. Most sweepers reside in the width range of 2 to 2.5 m. Their sweeping brushes are usually placed laterally and can cover a total street space from 2.5 to 3 m. Thus, our camera shall cover at least that width ($h_{wid}$), and horizontal field of view shall be derived as a consequence.

In order to choose the right vertical field of view, we will need to define $d_{near}$ as the distance (in meters) between the nearest pixel framed by the camera and the camera itself. Similarly, $d_{far}$ is the distance of the farthest pixel framed (Figure 2).

Given an expected *FPS* (frames per second) for the system (more on that in Section 3.3) and a maximum vehicle speed $v_{max}$, the first constraint to impose is the following:

$$d_{far} - d_{near} > \frac{v_{max}}{FPS}$$

to have a full coverage of the road ahead while the vehicle is moving. Most sweeper trucks are quite slow if compared to a normal car; in fact, peek speeds during cleaning operations may be in the range of 10 to 20 Km/h (about 3 to 5 m/s).

Thus, if we aim to have a system that analyzes the street at 1 frame per second, we will need to cover at least 5 m of ahead "vertical" ground to analyze the road with continuity.

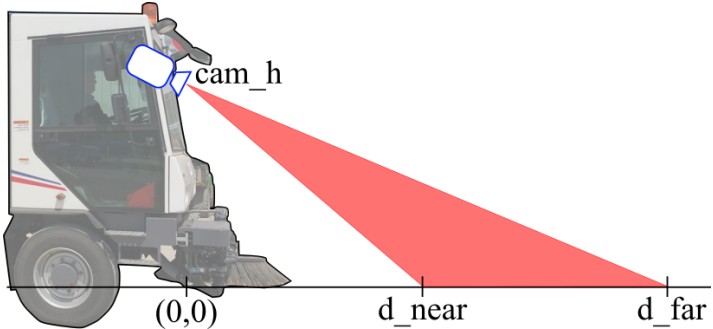

**Figure 2.** The "vertical" FOV of the system camera.

Another important consideration for camera placement is the minimum distance we are targeting with our camera ($d_{near}$). The choice of this distance will present some performance tradeoffs. By choosing a small $d_{near}$ we will have:

- A more responsive system. If the camera is framing near ground, the system reacts faster during quick turns of the truck;
- Less perceived kinetic speed when the truck is turning (the vision point of view is turning with a smaller radius), thus less blur when turning;
- Bumps and asperities will have a lower impact on the 3D world coordinates reconstruction error;
- Less subject to fog and rain;
- Better depth of field (DOF). A known effect with cameras is that the depth of field (the z-coordinate range in which the objects appear on focus) is larger when framing near objects.

While choosing a large $d_{near}$ we will have:

- More time to analyze the picture before having to act on the actuators (the objects to be analyzed are farther, so the vehicle takes more time to reach them, and the same goes for the deadline to the analysis to be completed. We called this concept *maximum latency of analysis*);
- Less blur when going straight (when moving straight and looking to near objects in perspective, they appear to move faster than far objects);
- Better perceived height for framed objects. Looking to a non-flat kind of trash (such as a bottle) is easier from a grazing angle than a top view.

In practice, the only *hard* constraint is the maximum allowed time to analyze a single frame ($t_{frame}$). In fact, the following must hold:

$$d_{near} > v_{max} \cdot t_{frame}$$

For simplicity, we impose $t_{frame} = 1/FPS$ (even if $t_{frame}$, the *maximum* allowed latency, could be higher than $1/FPS$, the *average* time of analysis for a frame in the system).

With these conditions in mind, and with the additional constraint of a camera placed inside the cockpit at a height from the ground called $cam_h$, which can be anywhere from 2 to 3 m, useful equations can be derived from simple geometrical equations. The vertical camera angle of view is:

$$\alpha_v = atan(\frac{d_{far}}{cam_h}) - atan(\frac{d_{near}}{cam_h})$$

while the horizontal angle of view is:

$$\alpha_h = 2 \cdot atan\left(\frac{h_{wid}}{2 \cdot \sqrt{d_{near}^2 + cam_h^2}}\right)$$

Using the approximate values of $d_{near} = 6$, $d_{far} = 13$, $cam_h = 3$, $h_{wid} = 3$ (expressed in meters), we get the following:

$$\alpha_h \approx 25°, \alpha_v \approx 14°$$

$d_{near}$ and $d_{far}$ have been chosen conservatively starting from the maximum vehicle speed of 5 m/s and targeting a system running at 1 FPS. Still, we did not choose too high $d_{near}$ and $d_{far}$ in order to exploit the benefits for a "small" $d_{near}$ listed previously. The final optic choice for our system has been a 25 mm optic that, in combination with the chosen camera, will grant to our system an $\alpha_h$ of 25° and an $\alpha_v$ of 16°.

### 3.2. The Computer Board

We list the requirements for the on-truck computer board:

- Low power consumption;
- Fast GPU;
- CanBus interface;
- Resistant to bumps;
- Replaceable (as every industrial asset).

For these reasons, we chose an Nvidia Jetson TX2 for the task. The two mainly conflicting requirements for this choice are GPU and resistance to bumps. The first is required for fast neural networks inference tasks, the second is mandatory for any vehicle that could face road asperities, holes and speed bumps.

Unfortunately, most recent GPUs are fan cooled, as are any product with moving parts that are not resistant to hits. Still, a truck has shock absorbers, and, as long as a human is required to drive a road sweeper, we suppose that the GPU will not be subject to extreme hits.

### 3.3. Neural Network for Garbage Detection

Detecting garbage and trash on a road from pictures calls for semantic segmentation. This system has the additional requirement of real-time detection, so the problem becomes also an instance of real-time segmentation.

A well-known and simple neural network for semantic segmentation is the so-called U-Net [20]. While the original U-Net is not real-time, a reduced version of the U-Net can, in the present year (2020), segment a 1 Megapixel image in under a second on a mobile GPU (as we will show in Section 4).

The first idea for this system was performing an end-to-end image-to-image transformation, from the original picture taken by the camera to a heatmap highlighting each piece of trash. The approach is similar to the task of segmenting road views for vehicles and pedestrians (e.g., CityScapes [26]). Differently from other approaches, we choose to train our system to detect only two classes: garbage and clean road (binary classification). First, we created a dataset of input road scenes and their expected output ground truths (highlighting garbage in red, while depicting the background in blue). Then, we trained an U-Net expecting the network to learn to replicate that behavior—i.e., detecting trash in the pictures. As input we used the original, unchanged pictures from the camera (perspective views like Figure 3, that yielded segmentations like the one in Figure 4).

The approach proved to work quite well, but we noticed dimensionality inconsistencies (e.g., the same object would have been overly detected when near, and not detected at all when far). Moreover, after the garbage detection phase, an inverse perspective transform was necessary to convert the trash position from perspective pixel coordinates to orthogonal 3D world coordinates.

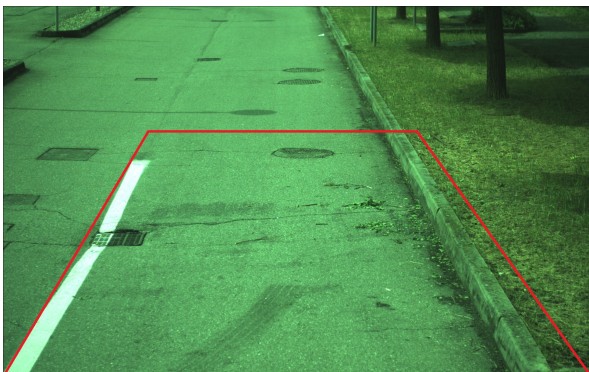

**Figure 3.** The original picture as taken by the camera (1920 × 1200 in our experiments), with highlighted the zone analyzed by algorithm.

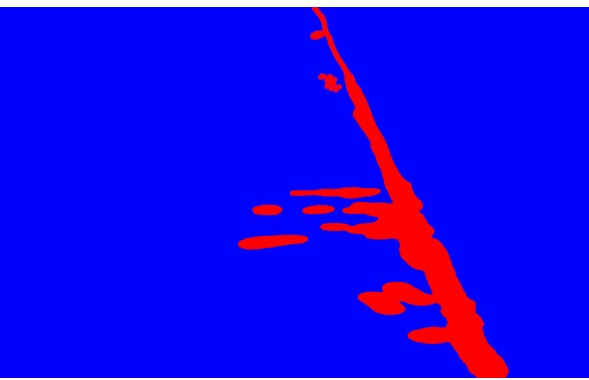

**Figure 4.** The semantic segmentation performed directly on the input (perspective) image (Figure 3). This approach is similar to the one of popular datasets like Cityscapes. Initial experiments with this setting showed its effectiveness, but the system has been further improved via orthogonal perspective transformation.

This convinced us to perform the inverse perspective transform directly on the input image, instead of on the results, and use that orthogonal, top-view image of the street for the network classification (since the road is assumed perfectly horizontal). See Figures 3 and 5 for an example of the transformation. Using this method, we were also immediately able to establish a precise pixels to millimeters correspondence that eased every consecutive operation.

The easiest way to obtain the perspective transformation matrix is to manually tape measure, in the real world, the four corner coordinates of the sweeper work area and highlight them in a camera picture (red boxes in Figures 3 and 5). These corner points will be mapped to:

$$(d_{near}, -\frac{h_{wid}}{2}), (d_{near}, \frac{h_{wid}}{2}), (d_{far}, -\frac{h_{wid}}{2}), (d_{far}, \frac{h_{wid}}{2})$$

by solving a linear system that yields the perspective transformation matrix.

Thus, using that matrix we will create a dataset of orthogonal birds-eye street views (Figure 5) that, after supplying their respective ground truth labels (Figure 6), we will use to train the network. The only remaining aspect to establish is a suitable pixel/millimeter ratio for our transformed pictures. The smallest piece of trash we want to detect is a cigarette stub, so we choose a sufficient ratio of 1:1 pixel/cm (i.e., one pixel corresponds to a 1 cm · 1 cm area in the real word). Higher resolutions would improve the quality of analysis but also notably slow down the performance of the system. This ratio will yield images of at least 300 × 700 pixels, given the aforementioned $d_{far} - d_{near} = 7$ and $h_{wid} = 3$ (in meters). In fact, we chose to stick with 500 × 700 images, having some additional lateral view in the higher part of the image due to perspective (the area outside the red box in Figure 5). This additional

framed area can help when the vehicle is turning, as we will see in Section 3.4, since we are already looking a bit on the right and on the left of the work area (one meter in both directions).

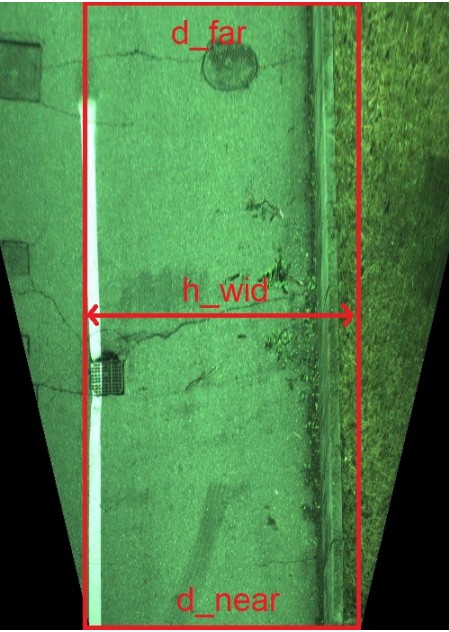

**Figure 5.** The orthogonal perspective transformation (@500×700), computed from the original Figure 3. Work area of the sweeper is highlighted. From this view, it is trivial to derive precise real world coordinates, given the scale of 1:1 pxl for cm. In the bottom of the picture ($d_{near}$) the visible area is limited to a width of $h_{wid}$, while in the upper part of the picture a larger road portion is visible.

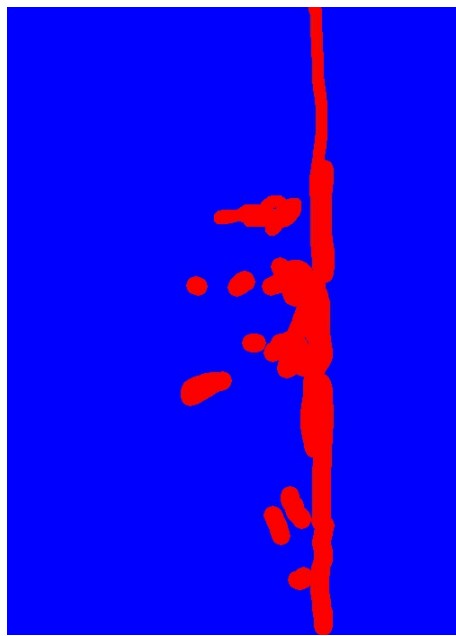

**Figure 6.** The ground truth picture (@500×700) after the perspective transformation. Blue is road background, red is trash and garbage.

3.3.1. Mini U-Net

Once we established an appropriate view and image size, the next step was developing a fast neural network for image semantic segmentation. We first tested the original U-Net topology defined by the paper with the same name. The traditional definition of U-Net proposes a downsampling layer (with factor two) to be added after every two consecutive $3 \times 3$ convolutions. Moreover, the topology in the paper doubles the number of image features (i.e., the channels of the convolutions) after each downsampling layer.

We ran experiments for the U-Net inference times on the Nvidia Jetson TX2, varying its architecture at each iteration, and found the following two main contributing factors to the execution times:

- Image resolution, with a linear relationship with processing time (this holds true around our resolution range; pictures with a higher or lower order of magnitude size could hit other performance limits: DRAM size, number of processing units of the hardware);
- Number of features in the bottleneck layers (the bottleneck layers of a network are the innermost layers, where the representation, and the picture, are smaller, but the number of features is larger).

This second factor somewhat surprised us, since we were expecting other factors to be more significant—e.g., the number of downsampling layers (with their reduction factors), and the total number of convolutional layers.

With these observations in mind, we redesigned the U-Net to obtain a faster Mini U-Net. We choose to stick with four downsampling steps (as the original paper) but we limited the number of features per layer to a maximum of 256, as opposed to the "full" U-Net which grew to 512, 1024 channels in the inner layers. With this improved topology, our Mini U-Net can run its inference in around 600 ms on a Nvidia Jetson TX2. The final network topology is the one depicted in Figure 7 and detailed in Table 1.

We should add that, to achieve these speeds, we also took advantage of TensorRT. TensorRT is a proprietary Nvidia SDK that works in a similar fashion as TensorFlow, and PyTorch, among others, but is further optimized for specific hardware, such as the Jetson.

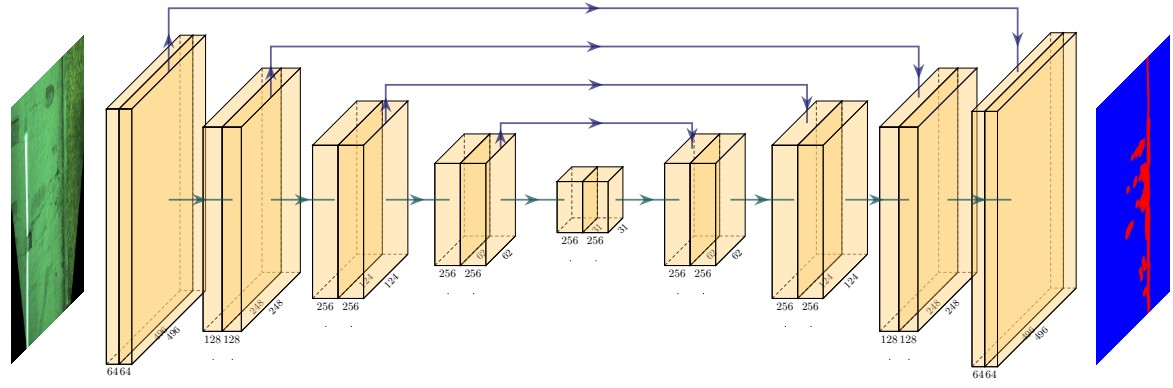

**Figure 7.** Our Mini U-Net for real-time dirt detection. Notably, the inner layers have just 256 features each.

Another consideration arises: the linearity between picture size and execution time could lead our system to use a smaller $d_{far}$ to obtain a smaller picture and thus higher FPS (without losing the full street coverage). This could allow us to use a different, more "pushed", camera and probably to have a more responsive system in general. This research line is left to be explored in future works.

**Table 1.** Our Mini U-Net topology. The 704 × 496 (input-output) dimension is the 16 nearest multiple dimension to the target (700 × 500) size. Notably only ≈8 M parameters are used, well below the training dataset size of 420 MB uncompressed (72 MB compressed). Every layer except the last uses ReLU as its activation function.

| Layer Type | Kernel | Features Size | Parameters |
|---|---|---|---|
| Input | | (704, 496, 3) | |
| Conv2d | (3, 3) | (704, 496, 64) | 1792 |
| Conv2d | (3, 3) | (704, 496, 64) | 36,928 |
| MaxPooling | (2, 2) | (352, 248, 64) | |
| Conv2d | (3, 3) | (352, 248, 128) | 73,856 |
| Conv2d | (3, 3) | (352, 248, 128) | 147,584 |
| MaxPooling | (2, 2) | (176, 124, 128) | |
| Conv2d | (3, 3) | (176, 124, 256) | 295,168 |
| Conv2d | (3, 3) | (176, 124, 256) | 590,080 |
| MaxPooling | (2, 2) | (88, 62, 256) | |
| Conv2d | (3, 3) | (88, 62, 256) | 590,080 |
| Conv2d | (3, 3) | (88, 62, 256) | 590,080 |
| MaxPooling | (2, 2) | (44, 31, 256) | |
| Conv2d | (3, 3) | (44, 31, 256) | 590,080 |
| Conv2d | (3, 3) | (44, 31, 256) | 590,080 |
| UpSampling | (2, 2) | (88, 62, 256) | |
| Conv2d | (3, 3) | (88, 62, 256) | 262,400 |
| Concatenate | | (88, 62, 512) | |
| Conv2d | (3, 3) | (88, 62, 256) | 1,179,904 |
| Conv2d | (3, 3) | (88, 62, 256) | 590,080 |
| UpSampling | (2, 2) | (176, 124, 256) | |
| Conv2d | (3, 3) | (176, 124, 256) | 262,400 |
| Concatenate | | (176, 124, 512) | |
| Conv2d | (3, 3) | (176, 124, 256) | 1,179,904 |
| Conv2d | (3, 3) | (176, 124, 256) | 590,080 |
| UpSampling | (2, 2) | (352, 248, 256) | |
| Conv2d | (3, 3) | (352, 248, 128) | 131,200 |
| Concatenate | | (352, 248, 256) | |
| Conv2d | (3, 3) | (352, 248, 128) | 295,040 |
| Conv2d | (3, 3) | (352, 248, 128) | 147,584 |
| UpSampling | (2, 2) | (704, 496, 128) | |
| Conv2d | (3, 3) | (704, 496, 64) | 32,832 |
| Concatenate | | (704, 496, 128) | |
| Conv2d | (3, 3) | (704, 496, 64) | 73,792 |
| Conv2d | (3, 3) | (704, 496, 64) | 36,928 |
| Conv2d | (3, 3) | (704, 496, 3) | 195 |
| | | | 8,288,067 |

### 3.3.2. The Dataset

We collected a dataset of hours of sweeper work to be used for the training. Conditions to be taken into account are: midday light, cloudy daylight, grazing sunset light, night with artificial lights.

A full range of road backgrounds should be taken into account for the artificial intelligence performance—e.g., asphalt vs. paving stones (see Figure 8).

We also framed most kinds of urban/natural trash, such as dirt, grass, leafs, bottles, plastic bags, cans, tissues, street market trash (rotten vegetables), with the results seen in Figure 9. Each of these kinds of garbage must be well-represented in the dataset in order to be found correctly during the inference.

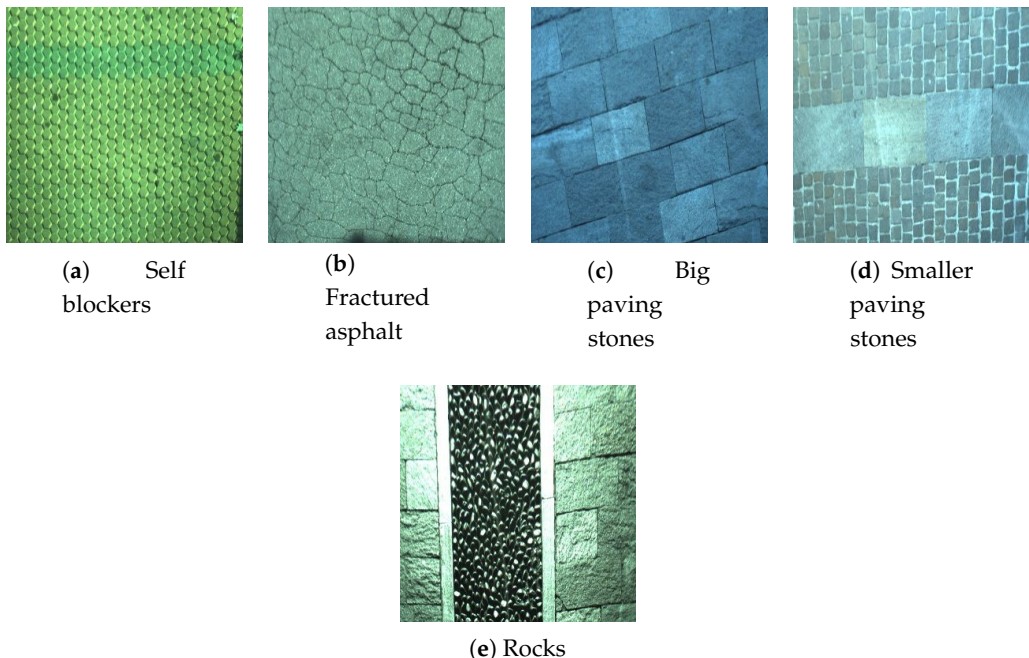

**Figure 8.** Some of the many backgrounds a road can exhibit (that we used in our training). They pose a great challenge for neural network generalization capabilities.

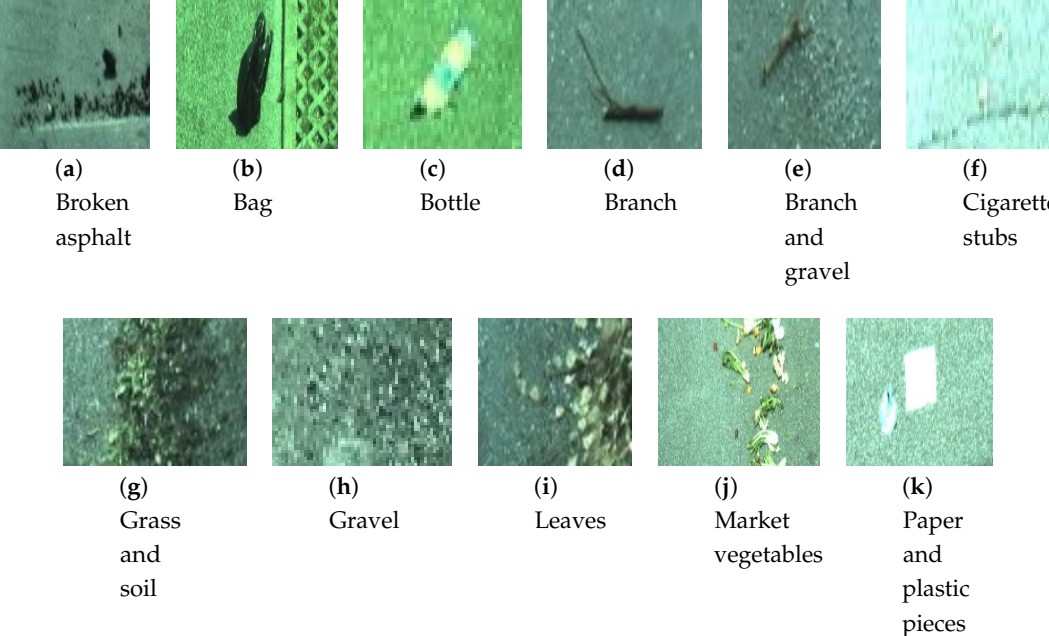

**Figure 9.** Typical kinds of urban trash to be detected by a cleaning system.

In particular, the dataset was created mounting a camera inside of one of those vehicles and "filming" more than 20 days of its work routine. This vehicle traveled around some northern Italian cities performing its normal collecting operations. Among these days, we picked a selection of daily routines that were as diverse as possible, in order to collect a representative dataset. Some of these routines were residential district clean-ups, industrial district clean-ups, cleaning routines performed in the zones of a weekly street market, and more. Each of these cleaning sections from different contexts lasted one to two hours, which we then sampled in order to compose a balanced set of data.

Moreover, we can consider some notable situations that the network could face. Figure 10 highlights some of these conditions. There are cases of bad image exposure, garbage that blends itself with the background and a priori knowledge that can be applied to help the network to learn. All of these situations should be accounted for and presented to the network during the training.

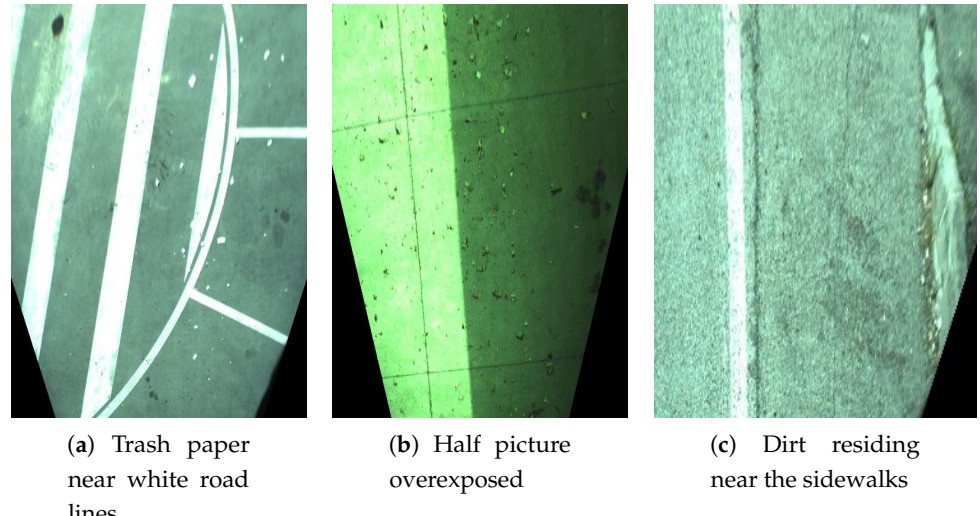

(**a**) Trash paper near white road lines

(**b**) Half picture overexposed

(**c**) Dirt residing near the sidewalks

**Figure 10.** Peculiar situations encountered during the dataset creation. (**a**) A hard condition for the network: it has to detect white pieces of paper near white road lines. The only way to do that is by learning their shapes (through example). (**b**)A poorly exposed image. Half of the picture is overexposed, while half underexposed. The network must be trained with such cases in order to work well during inference. Moreover, while it can learn to detect both overexposed and underexposed trash, the best solution to the problem would be equipping the system with a HDR camera. (**c**) A case of valuable a priori knowledge that can be exploited by the system. In fact, dirt usually gathers around nearby sidewalks. With this consideration in mind, the network can be trained to clean the sidewalks by default (by properly preparing the ground truth labels, such as in Figure 6), since finding a sidewalk is much easier than detecting each single cigarette stub.

Unfortunately, as a further demonstration that supervised learning has scalability issues, one must perform a full manual labeling of the whole dataset, providing the ground truth label for each of the street pictures. That is, for every input image to be processed (e.g., Figure 3), an expert human must create the relative ground truth picture (e.g., Figure 4), using the input picture as a guide, and over-painting the garbage zones in red while coloring the clean road zones in blue (something that can be done with any image processing software). Still, for the standards of neural networks, the Mini U-Net requires a relatively small number of examples, since we trained our system to a good degree of accuracy with as few as 400 image pairs (input + ground truth) in total (including as far as possible all of the aforementioned situations).

### 3.3.3. The Training

An important part of the training was *data augmentation*. Data augmentation is a technique performed before or during the training on the whole input dataset, in order to artificially augment the quantity of training samples. Typically, non-destructive image transformations are performed on the original picture to generate new images, which, while different, belong to the same classes as the originals. This is fundamental in overcoming the scalability issues typical of full supervised (thus limited in nature) datasets. The original dataset has been automatically rototranslated, zoomed and flipped with random parameters to simulate more conformations that the images could have and create a much bigger training set. Another important part of the data augmentation, that is usually overlooked in similar applications, is contrast and brightness augmentation. We altered our

samples reducing or enhancing the whole pictures contrast (multiplicative factor) and brightness (average value), in order to better recreate real world lighting scenarios.

The training was run using the Tensorflow library and the standard backpropagation—gradient descent algorithm. We tested Stochastic Gradient Descent, Adam [27] and Rmsprop. They gave similar results so we choose to use Adam with a learning rate of 0.0001. We ran our training for 1000 epochs.

One final consideration was the loss function for the net. Since the last layer of our network used softmax, the natural choice for a loss would be Categorical Cross-Entropy:

$$CCE = -\sum_{i}^{n}(y_{Ti} \cdot \log y_{Pi})$$

where $n$ is the number of output classes (two in our case), $y_T$ is the ground truth class (label) for each pixel of the image (encoded as one-hot), and $y_P$ is the predicted outcome from the softmax activation. However, we used a modified version of that loss, called Weighted Categorical Cross-Entropy:

$$WCCE = -\sum_{i}^{n}(y_{Ti} \cdot \log y_{Pi} \cdot w_i)$$

This kind of loss is just the usual cross-entropy, but with a per-class weight ($w$) used during the evaluation. These weights are constants defined a priori before the training. The application of these weights provides two important benefits:

1.  In the case of an unbalanced dataset, we can adjust the balance with just a multiplier (e.g., if we have more samples of good road than dirty road, we can increase the "dirt" class weight while keeping the other weight fixed—e.g., $w_{dirt} = 10, w_{clean} = 1$).
2.  We may want to train a more "aggressive" or "gentle" network. A customer may want to find 100% of trash even at the cost of producing some false positives; another customer could want to save energy and brushes and be tolerant of some gravel remaining uncleaned. With these weights, we can tune the same dataset towards a more, or less, aggressive network easily. The same could be done by changing the output threshold of the softmax output per-class, but in fact it did not work as well in our experiments: some tuning was possible but to a limited degree. To really change the network behavior towards a class or another, the choice has to be done before (and applied to) the training.

### 3.4. Real-Time Vehicle Control

The last part of the system is deploying a real-time functionality able to control the sweeper actuators via CanBus.

For this experiment, we are piloting a sweeper with just two brushes, one at the left and one at the right side of the vehicle. In the center of the vehicle, a belt is collecting all the garbage. Given these specifications, we need to accurately use the trash $x$ coordinate in order to activate the left, right and/or center actuators (our test truck has two lateral brushes and a central vacuum able to collect trash once lifted by the brushes).

Another important aspect to take into account is the $y$ coordinate of trash (i.e., the distance to the trash), that directly influences the latency before activating the brushes. Supposing that the vehicle is driving straight, at time $t_0$ we are framing garbage at a $y$ coordinate between $d_{near}$ and $d_{far}$. We need to delay that diagnosis and make the brushes spin only at a later time $t_1 = \frac{v_{avg}}{y}$, where $v_{avg}$ is the average vehicle speed during $[t_0, t_1]$ time interval. In fact, it is easier to reason with "space" instead of "time" and keep in memory a 2D buffer with real world trash $(x, y)$ coordinates, and make it "move" inside a simulation at the same speed of the vehicle. In that way we can emit cleaning commands when the vehicle finally encounters trash, with precision.

This could be done just by using the feedback of vehicle speed (which can be communicated by the vehicle itself via CanBus), or it could be further improved using the vehicle steering wheel

angle. With the steering wheel we could also rotate the 2D buffer of garbage locations taking into account vehicle turns. This would be critical for higher speed vehicles, but in our application vehicles were usually cleaning long, straight roads, so it is less mandatory. By using the steering wheel angle, and deploying a rotating diagnoses buffer, we can also take advantage of the wider lateral field of view in the upper part of the image, as introduced in Section 3.3. So, to sum up, with the vehicle speed and angle, we will navigate in the 2D simulated garbage buffer, and emit commands precisely when passing over trash.

Of course, in order to have a time accurate system, we have to place great attention to timing details. For vehicle speeds and wheel angles, we need to keep a rolling array with the last $n$ readings. These arrays must cover at least a time $t_{len} = \frac{d_{far}}{v_{max}}$ to not lose any needed past reading. So $n \geq \frac{refresh}{t_{len}}$ must hold, where $refresh$ is the refresh rate of the speed and steer that can be communicated via CanBus (the higher the better). As a last note, the system must use for each diagnosis the time point of when each picture was originally taken, not the time point of the end of the diagnosis.

## 4. Experimental Results

The system has been deployed in the real world, and has proven its correctness. The system has been tested for days of runtime on-board a sweeper truck. It has kept a steady $FPS = 1$ during all the tests, with no sign of slowdown or overheat. The $y$ accuracy of the system is considerably good, even if only the vehicle speed feedback has been used for the moving 2D garbage buffer (and not the aforementioned steering angle). When driving straight, the $y$ accuracy of the system is in the order of some centimeters with a CanBus vehicle refresh rate of 20 Hz.

The accuracy of the network, and the whole work, clearly depends on the quality and representative nature of the training dataset, but can be finely tuned with the weights of the Categorical Cross-Entropy. In our experiments with a dataset of over 400 images for training and about 80 testing images, the sensitivity and specificity of the system were 91% and 94%, respectively.

It is worth noting that power saving efficiency is directly proportional with the specificity of analysis, while the effective cleanup quality is directly related to the sensitivity term. It is also important to note that the test set (over which the accuracy is computed) has been purposely selected as hard cases for the network (partially over/under exposed pictures, hard backgrounds, wrong 3D perspective transformations). This has been done both to keep the better quality pictures for the training set (pictures which, is worth remembering, are limited in quantity since they need to be manually annotated) and also to highlight the performances for a "worst case scenario" for the network.

As weights for categorical cross entropy, we used $w_{dirt} = 13$, $w_{clean} = 1$ since in most real street pictures, even dirty ones, garbage is under-represented (covers at most 10% of the picture area). Figure 11 shows an ROC curve that we created training different networks with different $w_{dirt}$ parameters. This curve guided us in choosing the right $w_{dirt}$ (the one that maximizes Sensitivity + Specificity). From the curve, it is evident that not using a weighted categorical cross-entropy ($w_{dirt} = 1$) would greatly harm the accuracy of the selection process, in particular the Sensitivity term. This is another often overlooked aspect of training a neural network with an unbalanced dataset.

In Table 2 we also computed the metrics of IOU and DICE . These metrics, typically used for segmentation tasks, describe the ability of the network to correctly classify the class of *dirt* with spatial awareness. In general, an IOU above 50% can be considered good, and we note that this percentage is well reached by our system. From the table we can see that using a lower $w_{dirt}$ could further increase these metrics, at the price of a degraded sensitivity. For this application, we claim that sensitivity is more important, since IOU and DICE are heavily influenced by false positives, which are less of an issue in our domain: a few false positive percentage points only cause the roads to be "overcleaned" by that small percentage.

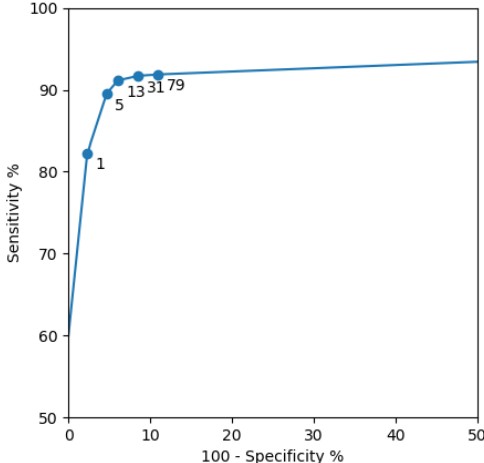

**Figure 11.** ROC curve (top-left quadrant) of our Mini U-Net for garbage detection. Each bullet represents a different network trained with that specific $w_{dirt}$ parameter. Other hyperparameters: lr = 0.0001, epochs = 1000, bottleneck layer number of features = 256.

**Table 2.** IOU and DICE metrics computed training the network with different $w_{dirt}$ weights. Other hyperparameters: lr = 0.0001, epochs = 1000, bottleneck layer number of features = 256.

| $w_{dirt}$ | IOU | DICE |
|---|---|---|
| 1 | 70.04% | 82.38% |
| 5 | 64.15% | 78.16% |
| 13 | 59.67% | 74.74% |
| 31 | 51.78% | 68.23% |
| 79 | 45.58% | 62.62% |

We also run tests to demonstrate the learning capabilities of the network with different topologies, proving that our Mini U-Net represents a good trade-off of inference time and representation power. The topology has been changed only for the three innermost (bottleneck) layers, respecting the U-shape typical of this architecture, and imposing a limit to the innermost features of 512 and 1024, respectively (see Figure 12).

The first test we run determined how the learning capabilities of the network change by varying its topology. In particular, we tested the two most relevant aspects for a cleaning scenario: specificity and sensitivity—see Table 3 for the results. The 256-features network exhibits performance very similar to the 512-sized one, which, surprisingly, is even better than the one with 1024 bottleneck features (the full, traditional, U-Net topology). Of course a bigger training dataset could, and should, influence these results, since in our case the complete network is probably overfitting.

The second test we run determined how the inference time of the network is influenced by its topology. Results are shown in Table 4 (tests run on a Jetson TX2 with TensorRT). While all these (average) times are under the theoretical limits for the system (1 FPS), the choice that has been made is to use the fastest topology (256 features and an inference time of 660 ms). This allows the system to be more resilient to its real-time constraints (a single spike in GPU execution time could make the system lose its pace and have it deadlock or lag behind the schedule). Moreover, the time saved by a faster inference can, and will, be used to perform other secondary, but important, operations, such as blob extraction and 3D reprojection.

The last test defined how fast each network trains, given its architecture—see Table 5 for results. While the increase in time seems to be sub-linearly related with the number of bottleneck features, still, it is quite relevant in absolute terms. Tests were run with a NVIDIA GeForce GTX 1070.

For all of these experiments, the hyperparameters were: $w_{dirt}$ = 13, lr = 0.0001, epochs = 1000.

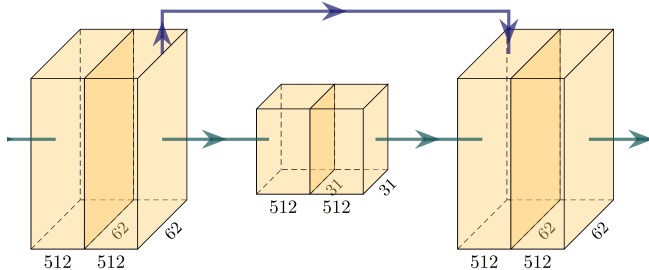

(**a**) 512 features bottleneck

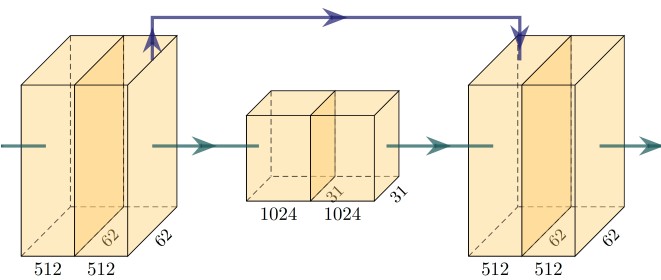

(**b**) 1024 features bottleneck

**Figure 12.** The altered networks run in our tests for representation power and speed.

**Table 3.** Learning capabilities for different-sized Mini U-Nets for garbage detection.

| N. of Bottleneck Features | Sensitivity | Specificity |
| --- | --- | --- |
| 256 | 91.14% | 93.96% |
| 512 | 91.31% | 95.19% |
| 1024 | 91.35% | 94.82% |

**Table 4.** Inference times for different-sized Mini U-Nets for garbage detection.

| N. of Bottleneck Features | Inference Time |
| --- | --- |
| 256 | 660 ms |
| 512 | 820 ms |
| 1024 | 930 ms |

**Table 5.** Training times for different-sized Mini U-Nets for garbage detection.

| N. of Bottleneck Features | Training Time Per Epoch |
| --- | --- |
| 256 | 119s |
| 512 | 144s |
| 1024 | 160s |

## 5. Conclusions

We presented a full system of artificial intelligence able to highly contribute to the working routine of road sweepers. The system may have a big impact in the domain of these vehicles, both from an economic and a ecological standpoint.

The system is reasonably cheap and easy to install/maintain, and has proven its correctness with multiple tests, showing its high accuracy and the robustness of its behavior.

Moreover, to the best of our knowledge, this is the first paper proposing such a real-time, hard-outdoor, operating solution for road vehicles. While other works may tackle similar domain toy examples, or, conversely, high cost solutions, this is the only work that provides a real world, fully integrated solution that can be applied with ease "in the wild" to gain immediate benefits.

## 6. Future Work

While this system has shown to be working fairly well for this task, a consecutive improvement would be training an end-to-end network. The system has been trained to detect one single class of "trash" and to collect it when present, from images. As we reported in the introduction, these machines have many cleaning facilities, and each facility is analogical in nature (cleaning pressure, brushes rotation speeds).

Modern neural networks can be trained end-to-end to learn these functions that map a picture of a road to the sweeper commands to clean that road. In that case, as a dataset, we would need both pictures, and the recording of each cleaning action performed by an expert sweeper driver.

## 7. Patents

- (International) Automated Road Sweeper and Road Cleaning Method using said road sweeper. WO 2020/152526 A1. 30/07/2020. Fabrizio Tagliaferri. Dulevo International Spa
- (Italian) Spazzatrice stradale automatizzata e procedimento di pulizia stradale di detta spazzatrice stradale. N. 102020000017542. 20/07/2020. Fabrizio Tagliaferri. Dulevo International Spa

**Author Contributions:** Conceptualization, L.D., F.T. and A.P.; Data curation, L.D.; Formal analysis, L.D.; Funding acquisition, A.P.; Investigation, L.D., F.T. and A.P.; Methodology, L.D. and F.T.; Project administration, A.P.; Resources, F.T. and A.P.; Software, L.D. and T.F.; Supervision, A.P.; Validation, L.D., T.F. and F.T.; Visualization, L.D. and T.F.; Writing—original draft, L.D.; Writing—review and editing, L.D., T.F., F.T. and A.P. All authors have read and agreed to the published version of the manuscript.

**Funding:** This work is funded by Dulevo International S.p.a.

**Acknowledgments:** We are really thankful to Dulevo for this opportunity.

**Conflicts of Interest:** The authors declare no conflict of interest.

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
