# Peer review of "An Energy Saving Road Sweeper Using Deep Vision for Garbage Detection"

_applsci, doi:10.3390/app10228146_

Round 1

Reviewer 1 Report

This paper presents a deep learning based method for garbage detection for road sweepers.

The issue is somewhat rarely studied until now, so the topic seems of interest to some group of people with new introduction of problem definition.

However, the paper needs to be improved in the following sense:

1) The target of detection is not clearly defined. What to detect in the scene should be clearly stated.

2) The data preparation is not explained in detail. What kinds of dataset it collected should be given in detail with figures and the labelling or ground truths.

3) How the dataset is composed of should be described in detail. How many images or videos it contains or the resolution/size and other details are not given.

4) How the result should be evaluated and how it should be compared to existing works have to be explained in detail.

In overall, the problem is not fully and formally defined and the target is not clear.

The constraints on the experiments and conditions for evaluations should be clearly stated.

Reviewer 2 Report

The paper presented an interesting road sweeper system that can automatically locate most types of road waste and instruct the truck to choose the proper operation. The waste detection task was handled using mini-Unet, a simplified version of Unet, in order to satisfy the real-time processing efficiency request. The whole system is well designed and tested using real-world data. The paper presented solid and novel work with a good application of deep learning techniques to a real-world new problem. My only concern is about the evaluation of the system. The authors may need to explain more in detail about the experiment result section. In details, my questions are:  

  1. Is the system able to distinguish different types of trash? In other words, when labeling the database, are the masks binary or with multi-class labels (for example, dirt and branch are labeled with different colors)?
  2. The output of the mini-Unet is a segmentation result, but the evaluation metrics (ROC, sensitivity, specificity) are for classification problems. Why the segmentation result is not evaluated using metrics such as DICE, IOU, but using classification metrics? If the segmentation result is analyzed at the pixel level, it is possible to view it as a classification problem, i.e., if each pixel is correctly classified into object or background, however, this pixel level classification evaluation cannot provide direct information about how well the segmentation is done. In short, a segmentation problem should be evaluated using metrics for the segmentation problem.
